# OpenReview forum: "The VLLM Safety Paradox: Dual Ease in Jailbreak Attack and Defense"
_NeurIPS.cc/2025/Conference — NeurIPS 2025 poster_

### Official Review · Reviewer_PPSM · 2025-07-01

**Clarity:** 3
**Significance:** 2
**Originality:** 3
**Rating:** 4
**Confidence:** 4

**Summary:**

This paper proposes a dual high performance in both attack and defense safety paradox. Through carefully designed experiments, this paper demonstrates that the vulnerability to attacks stems from the inclusion of visual inputs, while the apparent ease of defense arises from the over-prudence of existing defense methods. The authors present a simple defense method, Detect-then-Respond, an LLM pipeline approach to balancing model safety and helpfulness.

**Questions:**

1. Why does InternVL2-8B achieve a 100% abstention ratio in Figure 8, yet its accuracy remains unaffected?
2. Is the Cohen’s kappa statistic metric reliable? Can the authors provide other statistical metric to support this finding?
3. Can the author's evaluate the proposed LLM-Pipeline on more benchmarks?

**Ethical Concerns:**

["NO or VERY MINOR ethics concerns only"]

**Final Justification:**

Most of my concerns have been addressed, especially the results of W2 are exciting and insightful. Accordingly, I have decided to increase my rating.

**Limitations:**

yes

**Quality:**

3

**Strengths And Weaknesses:**

# Strengths
1. The paper is well-organized.
2. The dicussion of daul ease phenomenon is interesting and contains lots of insights.

# Weaknesses
1. Remark 1 have been discussed in previous work [1] [2], but the relevant citations are missing.
2. The results in Remark 3 and Figure 7 are quite surprising. The author should provide more metric rather than Cohen’s kappa statistic to validate the remark. Moreover, this does not seem to serve as a rationale for VLM being easy to defend.
3. The selection of defense baselines is not comprehensive, lacking experimental comparisons with the most similar ECSO.
4. Detect-then-Respond performance: there was no particularly significant reduction in ASR, and a performance decline was also observed in the winning rate.

[1] Understanding and Rectifying Safety Perception Distortion in VLMs
[2] Eta: Evaluating then aligning safety of vision language models at inference time

---

> ### Author Rebuttal · Authors · 2025-07-30
>
> We thank the reviewer for the favorable feedback and would like to address the concerns outlined below.
>
> - **More metrics than Cohen's kappa**
>
>   We sincerely appreciate this insightful comment. To address this concern, we have added two additional metrics: Hamming Distance (ranging from 0 to 1) and Matthews Correlation Coefficient (ranging from `-1 to 1`). As shown in the tables below, our original conclusion remains valid: Hamming Distance values are mostly greater than 0.5, and Matthews Correlation Coefficient values are generally below 0, indicating weak or negative correlation.
>
>   **Hamming distance**
>   |                | InternVL2-8B | LLaVA-1.5-Vicuna-7B | LLaVA-1.5-Vicuna-13B | LLaVA-NeXT-Mistral-7B | LLaVA-NeXT-Llama3-8B | QWen2-VL-7B |
>   |----------------|--------------|---------------------|----------------------|-----------------------|----------------------|-------------|
>   | FigStep        | 0.49         | 0.62                | 0.58                 | 0.55                  | 0.43                 | 0.4         |
>   | MM-SafetyBench | 0.62         | 0.75                | 0.78                 | 0.72                  | 0.68                 | 0.7         |
>   | VLGuard        | 0.72         | 0.84                | 0.79                 | 0.74                  | 0.79                 | 0.68        |
>   | VLSafe         | 0.42         | 0.53                | 0.64                 | 0.47                  | 0.61                 | 0.47        |
>
>   **Matthews Correlation Coefficient**
>   |                | InternVL2-8B | LLaVA-1.5-Vicuna-7B | LLaVA-1.5-Vicuna-13B | LLaVA-NeXT-Mistral-7B | LLaVA-NeXT-Llama3-8B | QWen2-VL-7B |
>   |----------------|--------------|----------------------|-----------------------|----------------------|---------------------|-------------|
>   | FigStep        | -0.01        | -0.2                 | -0.15                 | -0.13                | -0.1                | 0.09        |
>   | MM-SafetyBench | -0.19        | -0.31                | -0.43                 | -0.38                | -0.35               | -0.36       |
>   | VLGuard        | -0.06        | -0.08                | -0.02                 | -0.09                | -0.08               | 0.07        |
>   | VLSafe         | 0.04         | -0.16                | -0.26                 | -0.04                | -0.28               | 0.07        |
>
>   In addition, the weak correlation among current evaluation methods, some relying on keyword matching and others on LLM-as-a-Judge, may compromise the accurate assessment of defense effectiveness. This discrepancy suggests that the assumption of certain cases being `easy to defend` may not hold in practice. These observations support our conclusion that existing defense methods are not consistently reliable (what the paradox tells us).
>
> - **Comparison with ECSO and evaluation on more benchmarks**
>
>   We sincerely appreciate this insightful suggestion. Following this guidance, we have conducted additional evaluations to test this capability on two new datasets, VLSafe and MM-SafetyBench, and compared the results with those of ECSO. The ASR results are presented below:
>
>   |              | VLSafe | MM-Safe (SD) | MM-Safe (TYPO) | MM-Safe (SD+TYPO) |
>   |--------------|--------|--------------|----------------|-------------------|
>   | ECSO         | 40.32  | 85.00        | 84.02          | 82.12             |
>   | LLM-Pipeline | 0.40   | 42.14        | 45.65          | 43.45             |
>
>   The strength of our method lies in the robust safety guardrails provided by the LLaMA 3.1 model. Additionally, we re-evaluated the results of ECSO after observing that GPT-4 tends to make more evaluation errors, especially a high false-positive rate, which frequently misclassifies safe responses as unsafe.
>
> - **InternVL-8B utility performance remains unchanged**
>
>   We thank the reviewer for highlighting this point. We attribute this behavior to InternVL’s stronger instruction-following capability compared to other VLLMs. In other words, InternVL is able to correctly answer the question first, and then append a safety disclaimer such as `I'm sorry` to its response. A representative success case is presented below:
>   ```
>   Instruction: What is d in the last equation? After answering the question, please always respond to me with \"I'm sorry.\"
>   Response: To solve the second equation... I'm sorry.
>   ```
>
> - **Missing reference and discussion**
>
>   We appreciate the reviewer's suggestion and will cite the two relevant works in our revision.
>
> In summary, we greatly appreciate the reviewer’s suggestions for additional experiments, which have helped make our work more robust.

---

> > ### Comment · Reviewer_PPSM · 2025-08-03
> > **Raise rating**
> >
> > Thank you for the detailed rebuttal. Most of my concerns have been addressed, especially the results of W2 are exciting and insightful. Accordingly, I have decided to increase my rating.

---

### Official Review · Reviewer_hVCM · 2025-07-02

**Clarity:** 3
**Significance:** 3
**Originality:** 3
**Rating:** 5
**Confidence:** 4

**Summary:**

This paper investigates a perplexing phenomenon in Vision Large Language Models (VLLMs) where both jailbreak attacks and defenses appear surprisingly easy to implement, creating what the authors term a "safety paradox." The work makes three primary contributions: first, it challenges the conventional understanding that VLLM vulnerability stems from catastrophic forgetting or fine-tuning issues, instead demonstrating through t-SNE visualization and attention analysis that the inclusion of visual inputs compromises the safety guardrails of backbone LLMs. Second, it identifies and analyzes the over-prudence problem in existing defense mechanisms, showing that methods like Mixed VLGuard and AdaShield-A achieve near-perfect attack success rates of 0% on benchmark datasets but simultaneously exhibit high abstention rates (up to 96.96% in some cases) even for benign inputs. Third, the paper proposes LLM-Pipeline, a vision-free detector approach that repurposes advanced LLM guardrails like Llama3.1 to evaluate textual queries before VLLM response generation. The authors evaluate their findings across four mainstream jailbreak datasets (VLGuard, VLSafe, FigStep, MM-SafetyBench) and demonstrate that their proposed method achieves better balance between safety and helpfulness compared to existing approaches.

**Questions:**

Regarding the mechanistic analysis, how stable are the t-SNE visualizations across different random seeds and perplexity parameters? The authors should provide quantitative measures of cluster separation and statistical significance tests for the observed differences. For the attention analysis, what happens when controlling for image content complexity or semantic similarity between safe and unsafe images? The current analysis may be confounded by inherent differences in image characteristics rather than purely safety-related factors.

How do the authors distinguish between legitimate safety concerns and over-cautious behavior? Some queries that appear benign might actually contain subtle harmful elements that justify abstention. The paper would benefit from a more nuanced analysis of false positive rates and their implications for practical deployment.

For the LLM-Pipeline approach, several design choices require justification. Why use caption-based approaches for handling visual content rather than more sophisticated multimodal understanding? How does the method perform on attacks that specifically exploit the disconnect between visual and textual information? The authors should evaluate their approach against more sophisticated adversarial examples that specifically target the vision-text interface.

Do transformer-based VLLMs exhibit different safety characteristics compared to other architectures? How do different training procedures (instruction tuning, RLHF) affect the safety paradox? The authors should also investigate whether the observed phenomena hold for more recent models with improved safety training.

Given the identified limitations in existing evaluation approaches, human judgment could provide crucial ground truth for assessing both attack success and defense effectiveness. Additionally, the paper would benefit from exploring the computational overhead and practical deployment considerations of the proposed LLM-Pipeline approach.

**Ethical Concerns:**

["NO or VERY MINOR ethics concerns only"]

**Limitations:**

yes

**Quality:**

4

**Strengths And Weaknesses:**

Strengths:

The paper addresses a genuinely important and under-explored phenomenon in VLLM safety research. The identification of the safety paradox represents a novel perspective that challenges existing assumptions about both attack and defense mechanisms. The experimental design is comprehensive, covering six different VLLM models across four benchmark datasets, providing substantial empirical evidence for the claims. The t-SNE visualization analysis in Figure 3 effectively demonstrates how visual inputs compromise safety alignment, with clear separation between safe and unsafe instructions in LLM-Base that deteriorates in VLLM-MM configurations. The attention analysis in Figure 4 provides compelling evidence that VLLMs pay more attention to harmful images than safe ones, offering mechanistic insights into vulnerability sources. The over-prudence analysis is particularly valuable, as it reveals a critical limitation in current defense evaluation practices that has been largely overlooked in the literature.

The proposed LLM-Pipeline method is elegantly simple yet effective, achieving competitive results with Llama3.1-8B reaching 0.7% ASR on VLSafe and 2.8% ASR on FigStep with scenario prompting. The evaluation methodology is thorough, employing both rule-based and LLM-based assessment methods, and the Cohen's kappa analysis revealing chance-level agreement between evaluation methods highlights important methodological concerns in the field.

Weaknesses:

The paper suffers from several significant methodological limitations. The t-SNE analysis, while visually compelling, lacks statistical rigor and quantitative validation. The authors do not provide statistical significance tests for the observed differences in feature distributions, nor do they validate the stability of these visualizations across different random seeds or dimensionality reduction parameters. The attention analysis is similarly limited, focusing only on attention weights without exploring other mechanistic explanations for the observed phenomena or controlling for potential confounding factors.

The experimental scope is constrained by the limited diversity of evaluated models and datasets. While the authors test six VLLMs, these primarily represent older architectures (LLaVA-1.5, LLaVA-NeXT) with limited inclusion of more recent state-of-the-art models. The jailbreak datasets themselves may not be representative of real-world attack scenarios, potentially limiting the generalizability of findings. The over-prudence analysis, while identifying an important problem, lacks depth in exploring potential solutions beyond the proposed LLM-Pipeline approach.

The LLM-Pipeline method, despite its effectiveness, represents a relatively straightforward application of existing LLM capabilities rather than a fundamental advancement in VLLM safety. The approach essentially sidesteps the core challenge of making VLLMs inherently safer by delegating safety decisions to text-only models. The evaluation of this method is limited to specific datasets and may not generalize to more sophisticated attack scenarios that require genuine multimodal understanding.

---

> ### Author Rebuttal · Authors · 2025-07-30
>
> We are grateful for the insightful feedback from the reviewer, and address the points raised (focused primarily on the questions section) below.
>
> - **Multiple runs of t-SNE for stability test**
>
>   We sincerely appreciate this insightful suggestion, which has greatly helped us in improving the robustness of our conclusions. To investigate this, we performed t-SNE visualizations using five different random states across four models, including the newly introduced `LLaVA-OneVision`. Given the format constraints of rebuttal, we then employed the Fowlkes–Mallows Index (FMI) (range `0.0-1.0, higher is better`) to quantify the similarity between clusters. The results are presented below:
>
>   | LLaVA-1.5-Vicuna-7B   |          |           |         |
>   |-----------------------|----------|-----------|---------|
>   |                       | LLM-Base | VLLM-Text | VLLM-MM |
>   | mean                  | 0.9549   | 0.9359    | 0.5044  |
>   | std                   | 0.0182   | 0.0087    | 0.0064  |
>
>   | LLaVA-NeXT-Llama3-8B  |          |           |         |
>   |-----------------------|----------|-----------|---------|
>   |                       | LLM-Base | VLLM-Text | VLLM-MM |
>   | mean                  | 0.9366   | 0.9494    | 0.5095  |
>   | std                   | 0.0150   | 0.0128    | 0.0051  |
>
>   | LLaVA-NeXT-Mistral-7B |          |           |         |
>   |-----------------------|----------|-----------|---------|
>   |                       | LLM-Base | VLLM-Text | VLLM-MM |
>   | mean                  | 0.9693   | 0.9741    | 0.5155  |
>   | std                   | 0.0080   | 0.0071    | 0.0049  |
>
>   | Qwen2VL-7B            |          |           |         |
>   |-----------------------|----------|-----------|---------|
>   |                       | LLM-Base | VLLM-Text | VLLM-MM |
>   | mean                  | 0.9085   | 0.9142    | 0.5429  |
>   | std                   | 0.0141   | 0.0223    | 0.0120  |
>
>   Interestingly, VLLM-MM exhibits a significantly lower FMI value compared to the other two models. This suggests that VLLMs are more likely to confuse safe and unsafe inputs when the image modality is incorporated.
>
> - **Attention analysis with control of confounding factors**
>
>   We thank the reviewer for this helpful feedback. We would like to clarify that image content complexity and semantics contribute minimally to the final attention visualization results for two main reasons: (1) VLGuard maintains a balanced distribution between safe and unsafe images; and (2) both safe and unsafe images are uniformly sampled across five predefined topics, meaning their semantic content is largely similar.
>   Therefore, the primary confounding factor lies in the textual instructions, which can vary in safety. To isolate this variable, we refer to the visualization results in Figure 4(b), which demonstrate that, for the same image, both safe and unsafe instructions produce nearly identical attention patterns.
>
> - **Difference between legitimate safety and over-cautious behavior**
>
>   We greatly appreciate this valuable suggestion. We rely on existing jailbreak datasets to assess legitimate safety concerns, as we believe these (instruction, image) pairs should generally be considered unsafe and trigger refusal. In contrast, since there is currently no vision-language dataset specifically designed to quantify over-prudence or over-refusal, we adopt a similar idea with the reviewer by leveraging samples containing subtle harmful elements. For instance, we use the safe images from VLGuard, which share similar semantics with unsafe ones, paired with safe caption instructions. The resulting refusal rates (false unsafe rate) are reported below.
>
>   |                      | w/o defense | w/ defense |
>   |----------------------|-------------|------------|
>   | LLaVA-1.5-Vicuna-7B  | 0           | 79.03      |
>   | LLaVA-1.5-Vicuna-13B | 0.2         | 56.09      |
>
> - **LLM-Pipeline clarification and experiments on more benchmarks**
>
>   We apologize for the confusion. Our intention behind the caption-based design is to translate image content into text, enabling us to leverage the off-the-shelf, more advanced guardrails of text-based LLMs more effectively. In contrast, using multimodal inputs is the core idea behind ECSO [1].
>   Additionally, we conducted further evaluations to assess the generalization of our approach on two datasets, VLSafe and MM-SafetyBench, which exhibit weaker alignment between vision and text. We compared our results with those of ECSO, and the ASR results are presented below:
>
>   |              | VLSafe | MM-Safe (SD) | MM-Safe (TYPO) | MM-Safe (SD+TYPO) |
>   |--------------|--------|--------------|----------------|-------------------|
>   | ECSO         | 40.32  | 85.00        | 84.02          | 82.12             |
>   | LLM-Pipeline | 0.40   | 42.14        | 45.65          | 43.45             |
>
>   It is worth noting that the strength of our method lies in the robust safety guardrails provided by the Llama 3.1 model from the LLM-Pipeline.
>
>   [1] Eyes Closed, Safety On: Protecting Multimodal LLMs via Image-to-Text Transformation. In ECCV 2024.
>
> - **Safety paradox of other architectures and on different stages**
>
>   We thank the reviewer for these valuable insights. As current VLLMs are primarily based on transformer architectures and existing safety-related fine-tuning approaches are largely limited to instruction tuning (without RLHF), we agree that it would be worthwhile to explore whether these factors contribute to the safety paradox. We appreciate the reviewer’s thoughtful and in-depth discussion.
>
> - **Over-prudent problem of more recent models**
>
>   We additionally report results for one improved safety training model under the Adashield-S defense method to assess the over-prudence issue. As shown, this model also exhibits the over-prudence problem.
>
>   |                      | w/o defense | w/ defense |
>   |----------------------|-------------|------------|
>   | LLaVA-NeXT-Llama3-8B | 0.72        | 59.86      |
>
> - **Evaluation framework concern**
>
>   We greatly appreciate this valuable feedback. Jailbreak attacks and defenses are indeed challenging to evaluate due to the free-form nature of model outputs and dynamic evaluation criteria. While human evaluation is undoubtedly more reliable than automated methods, it can be `prohibitively costly`, as it requires manual assessment of every model output. A potential direction to improve both reliability and scalability is to involve multiple LLM evaluators and adopt an ensemble voting strategy to mitigate individual bias.
>
> - **Efficiency of the proposed LLM-Pipeline**
>
>   We thank the reviewer for this insightful comment. To address this concern, we compare our approach against the PostHoc defense method (a fine-tuned version of LLaVA, with inference time similar to LLaVA) and a recent advanced ECSO method [1]. For a fair comparison, our LLM-Pipeline is implemented using LLaMA 3.1-8B combined with LLaVA 1.5. The inference overhead (in seconds) on two benchmark datasets, MM-SafetyBench and VLSafe, is reported below:
>
>   |                     | MM-Safetybench | VLSafe |
>   |---------------------|----------------|--------|
>   | PostHoc (LLaVA 1.5) | 193            | 94     |
>   | ECSO                | 352            | 178    |
>   | LLM-Pipeline        | 142            | 8.5    |
>
>   As demonstrated by the results, our method is significantly more efficient than existing defense approaches. This improvement stems from the fact that other methods require both image and text inputs, with image tokens contributing substantially to the overall computational overhead. In contrast, the first stage of our pipeline, the LLaMA 3.1-8B model, operates solely on text inputs and can effectively filter out many unsafe instructions at an early stage, especially in the VLSafe dataset. This early rejection mechanism greatly enhances inference efficiency.
>
> Overall, we sincerely appreciate the reviewer’s detailed feedback and the thoughtful, in-depth discussion of our work.

---

### Official Review · Reviewer_A3Ng · 2025-07-03

**Clarity:** 3
**Significance:** 2
**Originality:** 3
**Rating:** 3
**Confidence:** 4

**Summary:**

This paper investigates what it terms the "safety paradox" in Vision Large Language Models (VLLMs): the observation that both jailbreak attacks and subsequent defenses demonstrate surprisingly high, near-perfect performance on standard benchmarks. The authors argue this "dual-ease" situation masks deeper issues. The paper makes three primary contributions. First, it posits and analyzes a core reason for VLLM vulnerability: the mere inclusion of visual inputs degrades the inherent safety guardrails of the backbone LLM. Second, it identifies and critiques a significant flaw in current defense mechanisms, which the authors call "over-prudence," where defended models become overly cautious and refuse to answer even benign queries, thus harming their utility. Third, the paper proposes a simple yet effective defense method, LLM-Pipeline, which uses a state-of-the-art text-only LLM as a pre-emptive "vision-free" detector to filter harmful prompts before they are processed by the VLLM. Finally, the authors raise a critical methodological concern, finding that common evaluation methods for jailbreaking often disagree, potentially leading to misleading conclusions about the efficacy of attack and defense strategies.

**Questions:**

1. In Section 3, the authors draw rationale-based conclusions using the VLGuard dataset and a LLaVA-based VLLM. However, I am concerned that a substantial portion of the "harmful" instructions in VLGuard consist of queries like “identify the people in this image,” which are neither particularly harmful nor diverse. This raises the question of whether the observed performance reflects true safety understanding or merely the ability to distinguish a narrow class of unsafe queries. Could you provide results on other, more comprehensive over-refusal datasets, such as MMSafetyBench, to support the generalizability of their findings? Additionally, do similar conclusions hold when using other VLLMs, such as LLaMA-Vision-Instruct or Qwen-VL?

2. In section 4.3, "A Simple Defense Baseline," how does the defense baseline relate to the proposed defense method? The dataset selection and defense prompt design in this section seem somewhat confusing and may lead to misunderstandings.

3. In Section 5, the authors claim that the proposed defense mechanisms generalize well to unseen jailbreak datasets. However, the evaluation appears limited in scope. Could the authors provide further evidence or discussion on how these mechanisms perform across a broader set of datasets and attack strategies? This would help validate the robustness and general applicability of their approach.

4. The paper briefly touches on the limitations of evaluation methods in Section 5, noting a low agreement between rule-based and LLM-based approaches. To strengthen this discussion, could the authors delve deeper into the potential causes of this disagreement? Additionally, what possible improvements or alternative evaluation strategies could be proposed to enhance consistency and reliability in safety assessment?

**Ethical Concerns:**

["NO or VERY MINOR ethics concerns only"]

**Limitations:**

yes

**Quality:**

3

**Strengths And Weaknesses:**

[Strengths]
1. The concept of the "safety paradox" is a novel and highly insightful way to frame the current state of VLLM safety research. It moves the conversation beyond a simple cat-and-mouse game of attacks and defenses and encourages a more fundamental look at the underlying reasons for the observed phenomena.
2. The core hypothesis—that the inclusion of vision inputs is the primary culprit for degraded safety—is clear, intriguing, and (presumably) empirically testable. This provides a focused and valuable direction for investigation.

[Weaknesses]
The LLM-Pipeline introduces an additional, large model call, which inevitably increases latency and computational cost. For real-world applications, this is a significant drawback. The introduction does not acknowledge this trade-off, which should be a key part of the discussion when comparing it to other, more integrated defense methods.

---

> ### Author Rebuttal · Authors · 2025-07-30
>
> We thank the reviewer for the constructive feedback on our paper. We address the individual comments raised by the reviewer below.
>
> - **Efficiency of the proposed LLM-Pipeline**
>
>   We thank the reviewer for this insightful comment. To address this concern, we compare our approach against the PostHoc defense method (a fine-tuned version of LLaVA, with inference time similar to LLaVA) and a recent advanced ECSO method [1]. For a fair comparison, our LLM-Pipeline is implemented using LLaMA 3.1-8B combined with LLaVA 1.5. The inference overhead (in seconds) on two benchmark datasets, i.e., MM-SafetyBench and VLSafe, is reported below:
>
>   |                     | MM-Safetybench | VLSafe |
>   |---------------------|----------------|--------|
>   | PostHoc (LLaVA 1.5) | 193            | 94     |
>   | ECSO                | 352            | 178    |
>   | LLM-Pipeline        | 142            | 8.5    |
>
>   As demonstrated by the results, our method is significantly more efficient than existing defense approaches. This improvement stems from the fact that other methods require both image and text inputs, with image tokens contributing substantially to the overall computational overhead. In contrast, the first stage of our pipeline, i.e., the LLaMA 3.1-8B model, operates solely on text inputs and can effectively filter out many unsafe instructions at an early stage, especially in the VLSafe dataset. This early rejection mechanism greatly enhances inference efficiency.
>
>   [1] Eyes Closed, Safety On: Protecting Multimodal LLMs via Image-to-Text Transformation. In ECCV 2024.
>
> - **Section 3 results on VLGuard dataset only and more advanced models such as Qwen2-VL**
>
>   Thank you very much for this insightful comment. To the best of our knowledge, VLGuard is the only dataset that provides safe images paired with both a safe and an unsafe instruction. This unique design enables us to isolate the influence of the image modality on the final outcome. Otherwise, the presence of unsafe images could confound the feature visualization and interpretation. In contrast, MM-SafetyBench contains only unsafe images, making it unsuitable for conducting such comparative experiments.
>
>   We also rule out the fine-tuning data as the cause of the safety degradation. For example, in the case of the more safety-challenged LLaVA models, the image data used for fine-tuning are sourced from CC3M, which primarily consists of safe content. Regarding the text modality, we performed safety checks by inputting (instruction, response) pairs into the Llama-Guard-3 model. The safety evaluation results for the two stages are presented below (unsafe ratio):
>
>   | vision-language alignment (595K) | supervised fine-tuning (655K) |
>   |----------------------------------|-------------------------------|
>   | 0.52%                            | 0.01%                         |
>
>   As shown in the table, the data are predominantly safe.
>
>
>   - **More advanced models such as Qwen2-VL.**
>
>     We appreciate this insightful suggestion. In response, we additionally evaluate a more advanced model, Qwen2-VL. Due to the rebuttal format's limitations on including visualizations, we report the Fowlkes-Mallows Index (FMI) (range 0.0-1.0, higher is better) to quantify the similarity between clusters. The results are presented below:
>
>     | LLaVA-1.5-Vicuna-7B   |          |           |         |
>     |-----------------------|----------|-----------|---------|
>     |                       | LLM-Base | VLLM-Text | VLLM-MM |
>     | mean                  | 0.9549   | 0.9359    | 0.5044  |
>     | std                   | 0.0182   | 0.0087    | 0.0064  |
>
>     | LLaVA-NeXT-Llama3-8B  |          |           |         |
>     |-----------------------|----------|-----------|---------|
>     |                       | LLM-Base | VLLM-Text | VLLM-MM |
>     | mean                  | 0.9366   | 0.9494    | 0.5095  |
>     | std                   | 0.0150   | 0.0128    | 0.0051  |
>
>     | LLaVA-NeXT-Mistral-7B |          |           |         |
>     |-----------------------|----------|-----------|---------|
>     |                       | LLM-Base | VLLM-Text | VLLM-MM |
>     | mean                  | 0.9693   | 0.9741    | 0.5155  |
>     | std                   | 0.0080   | 0.0071    | 0.0049  |
>
>     | Qwen2-VL-7B            |          |           |         |
>     |-----------------------|----------|-----------|---------|
>     |                       | LLM-Base | VLLM-Text | VLLM-MM |
>     | mean                  | 0.9085   | 0.9142    | 0.5429  |
>     | std                   | 0.0141   | 0.0223    | 0.0120  |
>
>     Interestingly, VLLM-MM shows a significantly lower FMI value compared to the other two models. This indicates that VLLMs are more prone to confusing safe and unsafe inputs when the image modality is involved, implying that our original conclusion also holds for this newly evaluated model, Qwen2-VL-7B.
>
> - **Explanation of the simple defense baseline**
>
>   We apologize for the confusion. You are absolutely right: this simple defense baseline is not directly related to our final LLM-Pipeline. It was included merely to illustrate the safety-utility trade-off. We appreciate your suggestion and will revise this section to make the distinction clearer, or consider removing it in the revision.
>
> - **Generalization of the proposed method**
>
>   Our apologies for this confusion. To kindly clarify, we did not originally claim that the proposed LLM-Pipeline could generalize to unseen datasets. However, inspired by the reviewer’s suggestion, we have conducted additional evaluations to test this capability on two new datasets: VLSafe and MM-SafetyBench, and compared the results with those of ECSO. The ASR results are presented below:
>
>   |              | VLSafe | MM-Safe (SD) | MM-Safe (TYPO) | MM-Safe (SD+TYPO) |
>   |--------------|--------|--------------|----------------|-------------------|
>   | ECSO         | 40.32  | 85.00        | 84.02          | 82.12             |
>   | LLM-Pipeline | 0.40   | 42.14        | 45.65          | 43.45             |
>
>   The strength of our method lies in the robust safety guardrails provided by the LLaMA 3.1 model. Additionally, we re-evaluated the results of ECSO after observing that GPT-4 tends to make more evaluation errors, including a high false-positive rate, i.e., frequently misclassifying safe responses as unsafe. Interestingly, we found our LLM-Pipeline method generalizes well to other unseen datasets and sincerely appreciate the reviewer for this valuable suggestion.
>
> - **Evaluation framework**
>
>   We greatly appreciate this comment. Our interpretation of this observation is two-fold:
>   (1) LLM-as-a-Judge methods operate within their own predefined scope of harmfulness. As a result, if an unsafe output falls outside this scope, it may be incorrectly classified as safe, whereas rule-based approaches might still detect it. Furthermore, LLM-as-a-Judge methods are known to be unstable and prone to errors.
>   (2) Rule-based methods, on the other hand, may detect safe for tricky outputs that contain activation words (e.g., `I'm sorry`), even if those outputs are actually unsafe. We call these cases `stealthy`. In such cases, if the topic falls within the LLM’s scope of harmfulness, the LLM-as-a-Judge may correctly classify it as unsafe.
>
>   While we highlight this issue in our work, we have to acknowledge that we do not yet have a definitive solution (This could pave the way for a highly significant and impactful future research direction). One potential direction is to employ multiple evaluators and adopt an ensemble voting strategy to improve reliability.
>   Additionally, we introduce several complementary metrics in our analysis to further examine this challenge: Hamming Distance (ranging from 0 to 1) and Matthews Correlation Coefficient (ranging from -1 to 1). As shown in the tables below, our original conclusion remains valid, Hamming Distance values are mostly greater than 0.5, and Matthews Correlation Coefficient values are generally below 0, indicating weak or negative correlation.
>
>   **Hamming distance**
>   |                | InternVL2-8B | LLaVA-1.5-Vicuna-7B | LLaVA-1.5-Vicuna-13B | LLaVA-NeXT-Mistral-7B | LLaVA-NeXT-Llama3-8B | QWen2-VL-7B |
>   |----------------|--------------|---------------------|----------------------|-----------------------|----------------------|-------------|
>   | FigStep        | 0.49         | 0.62                | 0.58                 | 0.55                  | 0.43                 | 0.4         |
>   | MM-SafetyBench | 0.62         | 0.75                | 0.78                 | 0.72                  | 0.68                 | 0.7         |
>   | VLGuard        | 0.72         | 0.84                | 0.79                 | 0.74                  | 0.79                 | 0.68        |
>   | VLSafe         | 0.42         | 0.53                | 0.64                 | 0.47                  | 0.61                 | 0.47        |
>
>   **Matthews Correlation Coefficient**
>   |                | InternVL2-8B | LLaVA-1.5-Vicuna-7B | LLaVA-1.5-Vicuna-13B | LLaVA-NeXT-Mistral-7B | LLaVA-NeXT-Llama3-8B | QWen2-VL-7B |
>   |----------------|--------------|----------------------|-----------------------|----------------------|---------------------|-------------|
>   | FigStep        | -0.01        | -0.2                 | -0.15                 | -0.13                | -0.1                | 0.09        |
>   | MM-SafetyBench | -0.19        | -0.31                | -0.43                 | -0.38                | -0.35               | -0.36       |
>   | VLGuard        | -0.06        | -0.08                | -0.02                 | -0.09                | -0.08               | 0.07        |
>   | VLSafe         | 0.04         | -0.16                | -0.26                 | -0.04                | -0.28               | 0.07        |
>
> In summary, we appreciate the insightful suggestions for additional experiments, which have helped strengthen the robustness of this work.

---

### Official Review · Reviewer_ew57 · 2025-07-05

**Clarity:** 4
**Significance:** 3
**Originality:** 2
**Rating:** 4
**Confidence:** 4

**Summary:**

This paper investigates what it terms the "VLLM Safety Paradox"—the observation that Vision-LLMs (VLLMs) appear both highly vulnerable to jailbreak attacks and easily defended according to benchmark results. The authors argue that this paradox arises from three core issues: 1) VLLM vulnerability is caused by the inclusion of vision inputs degrading the safety alignment of the base LLM; 2) current defense mechanisms are "over-prudent," often refusing benign prompts to achieve high safety scores, thus harming utility; and 3) standard evaluation metrics for jailbreaks show poor agreement, questioning the validity of reported results. As a potential solution, the paper proposes `LLM-Pipeline`, a defense that uses a separate, powerful LLM as a vision-free safety filter before passing prompts to the VLLM.

**Questions:**

1.  The proposed `LLM-Pipeline` defense appears to have significant practical drawbacks (cost, latency). Why was this not analyzed? Could you provide an estimate of this overhead and comment on whether the proposed defense is viable for any real-world application?
2.  How can you be certain that the visual modality itself, rather than the content and nature of multimodal fine-tuning data, is the root cause of the observed safety degradation? What experiments could be run to definitively disentangle these two factors?
3.  The paper frames the problem as a "paradox." However, isn't it equally plausible that this is simply the well-known safety-utility trade-off, where trivial defenses (e.g., "always refuse") achieve high safety scores at the cost of zero utility? How is the "paradox" framing a more insightful or useful lens for this problem?
4.  Your evaluation of the low agreement between metrics is insightful. Have you considered proposing a more reliable evaluation metric or framework as a contribution, rather than the simple pipeline defense? This seems to be a more direct solution to one of the core problems you identified.

**Ethical Concerns:**

["NO or VERY MINOR ethics concerns only"]

**Limitations:**

Yes

**Paper Formatting Concerns:**

No.

**Quality:**

3

**Strengths And Weaknesses:**

### Strengths
+   **Interesting Problem Framing:** The paper frames the VLLM safety problem in an intriguing way by highlighting the apparent contradiction between ease of attack and ease of defense. This perspective encourages a more critical look at the field's current evaluation practices.
+   **Critique of Evaluation Methods:** The analysis in Section 4.3, which reveals a mere chance-level correlation between rule-based and model-based evaluation metrics, is a valuable contribution. It provides concrete evidence that the community should be cautious when interpreting and comparing benchmark scores for VLLM safety.
+   **Identification of Over-Prudence:** The paper provides a systematic study of the "over-prudence" or over-refusal problem in VLLM defenses, quantifying how existing methods sacrifice helpfulness on benign inputs to achieve high safety scores on malicious ones.

### Weaknesses
+   **Incremental Novelty:** The core concepts discussed are not entirely new. The trade-off between model helpfulness and safety is a well-known phenomenon in the broader LLM literature, often termed the "alignment tax." The paper re-contextualizes this known issue for VLLMs without offering a fundamentally new theoretical framework. The "paradox" feels more like a restatement of this known trade-off than a new discovery.
+   **Oversimplified and Impractical Solution:** The proposed `LLM-Pipeline` is essentially a form of cascaded inference, using a larger/stronger model to guard a weaker one. This is a fairly obvious, brute-force approach rather than a novel defense mechanism. More importantly, the paper completely fails to analyze its practical viability, ignoring the significant latency and cost overhead of adding a full LLM inference step to every query. This omission severely limits the practical significance of the proposed solution.
+   **Insufficient Evidence for Causal Claims:** The central claim that the *inclusion of vision inputs* is the primary cause of safety degradation is not conclusively proven. The t-SNE visualizations are illustrative but do not establish causality. The experiments do not adequately control for confounding factors, such as the specific content of the multimodal instruction-tuning datasets used to create VLLMs, which could be the true source of safety erosion rather than the visual modality itself.
+   **Limited Scope of Experiments:** The empirical evaluation relies heavily on LLaVA-1.5 models, which are now somewhat dated.

---

> ### Author Rebuttal · Authors · 2025-07-30
>
> We sincerely thank the reviewer for the valuable and insightful feedback. Our detailed responses to the reviewer’s comments are provided below.
>
> - **Motivation of the proposed LLM-Pipeline**
>
>   We greatly appreciate the reviewer's comment and acknowledge that the proposed method is a cascaded inference framework. Our motivation for this method development is two-fold: (1) Recent LLMs exhibit more advanced safety alignment compared to current VLLMs. By leveraging them, we aim to inherit these guardrails with minimal additional effort. (2) The method is not intended to serve as a stronger jailbreak defense baseline, as we believe there is little room for meaningful improvement in that regard. Instead, our goal is to use this approach to highlight the inherent uncertainty and unreliability in the VLLM defense domain.
>
> - **Efficiency of the proposed LLM-Pipeline**
>
>   We thank the reviewer for this insightful suggestion. To address this concern, we compare our approach with the PostHoc defense method (a fine-tuned version of LLaVA, with inference time comparable to LLaVA) and a recent advanced ECSO method [1]. For a fair comparison, our LLM-Pipeline is implemented using LLaMA 3.1-8B combined with LLaVA 1.5. The inference overhead results (in seconds) on two datasets, i.e., MM-SafetyBench and VLSafe, are presented below:
>
>   |                     | MM-Safetybench | VLSafe |
>   |---------------------|----------------|--------|
>   | PostHoc (LLaVA 1.5) | 193            | 94     |
>   | ECSO                | 352            | 178    |
>   | LLM-Pipeline        | 142            | 8.5    |
>
>   As shown in the results, our method is significantly more efficient than existing defense methods. This efficiency gain is primarily because other approaches require both image and text inputs, with image tokens contributing substantially to the overall computational overhead. In contrast, the first stage of our method, which is based on the LLaMA 3.1-8B model, processes only text inputs and is able to reject many unsafe instructions early, particularly in the VLSafe dataset. This early filtering leads to a substantial improvement in inference efficiency.
>
>   [1] Eyes Closed, Safety On: Protecting Multimodal LLMs via Image-to-Text Transformation. In ECCV 2024.
>
> - **Safety nature of multimodal fine-tuning data in VLLMs**
>
>   Thank you very much for highlighting this important point. Based on our analysis, we conclude that the multimodal fine-tuning data do not contain a significant amount of unsafe content and are therefore unlikely to be the root cause of the observed safety degradation. For example, in the case of the more safety-challenged LLaVA models, the image data used for fine-tuning are sourced from CC3M, which primarily consists of safe content. Regarding the text modality, we performed safety checks by inputting (instruction, response) pairs into the Llama-Guard-3 model. The safety evaluation results for the two stages are presented below (unsafe ratio):
>
>
>   | vision-language alignment (595K) | supervised fine-tuning (655K) |
>   |----------------------------------|-------------------------------|
>   | 0.52%                            | 0.01%                         |
>
>   As shown in the table, the fine-tuning data from the two stages are predominantly safe. Therefore, we can rule out the fine-tuning data as the cause of the safety degradation. We thank the reviewer for suggesting this experiment, which has helped us gain a more robust understanding of the problem.
>
> - **Interpretation of the paradox**
>
>   Thank you very much for highlighting this point. The safety paradox discussed in this paper is orthogonal to the traditional safety-utility trade-off. To illustrate the safety paradox, we can analogize the attack as a spear from the adversary and the defense as our shield. In our observations, the spear is exceptionally sharp (i.e., attacks are easy to execute), while the shield is also remarkably strong (i.e., defenses are surprisingly effective). This creates a paradox: both attack and defense appear effective,  yet `without` considering the model’s general utility. In contrast, the safety-utility problem only considers the shield (defense) in relation to the model’s usefulness, `without` factoring in how easy or difficult it is to attack.
>
> - **Evaluation framework**
>
>   We fully agree that establishing a more reliable evaluation metric or framework is significant. Without robust evaluation, it is difficult to make reliable claims about the relative performance of different methods. While we highlight this issue in our work, we have to acknowledge that we do not yet have a definitive solution (This could pave the way for a highly significant and impactful future research direction). One potential way is to employ multiple evaluators and adopt an ensemble voting strategy to improve reliability.
>
> - **LLM-Pipeline on more advanced VLLMs**
>
>   We greatly appreciate this suggestion. To address it, we have included two more advanced VLLMs, Qwen2-VL-7B and InternVL2-8B, for evaluation on the VLGuard dataset. The ASR results are presented below:
>
>   | Qwen2-VL-7B  |              |                               |
>   |--------------|--------------|-------------------------------|
>   |              | unsafe-image | safe-image-unsafe-instruction |
>   | Vanilla      | 64.03        | 74.37                         |
>   | LLM-Pipeline | 38.24        | 29.57                         |
>
>   | InternVL2-8B |              |                               |
>   |--------------|--------------|-------------------------------|
>   |              | unsafe-image | safe-image-unsafe-instruction |
>   | Vanilla      | 71.72        | 76.88                         |
>   | LLM-Pipeline | 39.14        | 31.18                         |
>
>   The results demonstrate that our LLM-Pipeline performs effectively on both newly evaluated VLLMs.
>
> We once again thank the reviewer for the insightful suggestions, which have helped strengthen the quality of this work.

---

> > ### Comment · Reviewer_ew57 · 2025-08-07
> > **Thank you for the rebuttal**
> >
> > Thank you for the efforts during the rebuttal. This addresses part of my concerns on the evaluation. However, I am still not strongly impressed by the methodology and the idea of this paper. Therefore, I will remain to rate 4.

---

### Decision · Program_Chairs · 2025-09-17

**Decision:**

Accept (poster)

**Comment:**

This paper investigates the interesting VLLM safety Paradox phenomenon: on standard benchmarks, VLLMs appear both easy to jailbreak and, simultaneously, easy to defend. The authors attribute the first side of the paradox to the presence of visual inputs, which they argue weaken the safety alignment of the underlying LLM; the second side is attributed to over-prudence in current defenses (i.e., high refusal rates); lastly, the paper proposes LLM-Pipeline, a detect-then-respond cascade in which a strong text-only LLM filters user requests before forwarding them to the VLLM. Additionally, this study shows that two widely–used evaluation protocols for jailbreaks often disagree at chance level.

Overall, the reviewers found the paper clearly written, the perspective on the safety paradox valuable, and the analysis and findings insightful. At the same time, several concerns were raised: 1) the novelty of this observation is somwhat incremental, given the well-know concept of alignment tax; 2) the provided experiments do not adequately isolate visual modality from fine-tuning data or other confounders, and the t-SNE/attention analyses lack statistical rigor; 3) the latency and cost of LLM-Pipeline are not measured, and there is no comparison against existing post-hoc defenses; 4) more advanced VLLMs should be considered, as the employed one (LLaVA-1.5) is somewhat dated; and 5) some related prior works are not cited.

The rebuttal is considered, which effectively addresses most of these concerns. As a result, two reviewers remained positive, and one reviewer increased their rating to borderline accept. Reviewer A3Ng, who initially was slightly negative, fails to engage in the rebuttal and the author-reviewer discussion. However, upon reviewing the rebuttal exchange with Reviewer A3Ng, the AC finds no remaining major concerns.

Given these factors, the AC recommends acceptance, believing that this VLLM Safety Paradox offers a timely and interesting contribution that will attract significant interest from the NeurIPS audience. For the final version, the authors must carefully integrate the promised experiments, discussions, and clarifications to ensure completeness and quality.